# CAML: Collaborative Auxiliary Modality Learning for Multi-Agent Systems

**Rui Liu[1], Yu Shen[2], Peng Gao[3], Pratap Tokekar[1], Ming Lin[1]**
[1]University of Maryland, College Park
[2]Adobe Research
[3]North Carolina State University

## Abstract

Multi-modal learning has emerged as a key technique for improving performance across domains such as autonomous driving, robotics, and reasoning. However, in certain scenarios, particularly in resource-constrained environments, some modalities available during training may be absent during inference. While existing frameworks effectively utilize multiple data sources during training and enable inference with reduced modalities, they are primarily designed for single-agent settings. This poses a critical limitation in dynamic environments such as connected autonomous vehicles (CAV), where incomplete data coverage can lead to decision-making blind spots. Conversely, some works explore multi-agent collaboration but without addressing missing modality at test time. To overcome these limitations, we propose Collaborative Auxiliary Modality Learning (**CAML**), a novel multi-modal multi-agent framework that enables agents to collaborate and share multi-modal data during training, while allowing inference with reduced modalities during testing. Experimental results in collaborative decision-making for CAV in accident-prone scenarios demonstrate that **CAML** achieves up to a $\mathbf{58.1}\%$ improvement in accident detection. Additionally, we validate **CAML** on real-world aerial-ground robot data for collaborative semantic segmentation, achieving up to a $\mathbf{10.6}\%$ improvement in mIoU.

## 1 Introduction

Multi-modal learning has become an essential approach in a wide range of machine learning systems, particularly in areas such as autonomous driving [6, 7, 27, 46], robotics [20, 26, 36], and reasoning [21, 28, 29], where the availability of multiple data sources (e.g., RGB, LiDAR, radar, text) improves model performance by providing complementary information. However, these multi-modal systems often incur increased computational overhead and latency at inference time. More critically, in real-world conditions, some modalities may be unavailable, or too costly to acquire at test time, which challenges the reliability and efficiency of such systems.

Recent work in machine learning [9, 10, 16, 37, 45] has sought to address these problems by enabling models to leverage additional modalities during training while supporting inference with fewer modalities. Such approaches reduce computational costs and improve robustness in conditions where some sensors may be unavailable at inference time. Shen et al. [41] formalized these tasks under the framework of Auxiliary Modality Learning (AML), which effectively reduces the dependency on expensive or unreliable modalities.

Despite the benefits of AML, notable limitations remain. AML is primarily designed for single-agent settings, where an individual model is trained to handle reduced modalities during inference. A key challenge in current AML frameworks is insufficient data coverage, particularly in dynamic environments such as *connected autonomous vehicles* (CAV). In these scenarios, a single agent's data

is often incomplete due to occlusions or limited sensor range, resulting in blind spots and increased uncertainty in decision-making. In contrast, multi-agent systems, enabled by technologies such as vehicle-to-vehicle (V2V) communication or collaborative robotics, can access complementary sensory information from other agents. Sharing this information allows for more informed and safer decisions, especially in accident-prone scenarios. However, existing single-agent AML approaches fail to leverage this collaborative potential.

To address these limitations, we propose Collaborative Auxiliary Modality Learning (**CAML**), a novel framework for multi-modal, multi-agent systems. **CAML** enables agents to collaborate and share multi-modal data during training while supporting inference under reduced-modality conditions at test time, as illustrated in Fig. 1. This collaborative learning paradigm enhances scene understanding and data coverage by allowing agents to compensate for each other's blind spots, leading to more informed and coordinated decision-making. To ensure robust deployment in scenarios with missing modalities (e.g., due to sensor failures or resource constraints), **CAML** employs knowledge distillation (KD) [15] to transfer knowledge from a teacher model trained with full-modality inputs to a student model that performs effectively with limited modalities. By enabling cross-modal and cross-agent knowledge transfer, **CAML** extends traditional KD to handle partial observability and support team-level coordina-

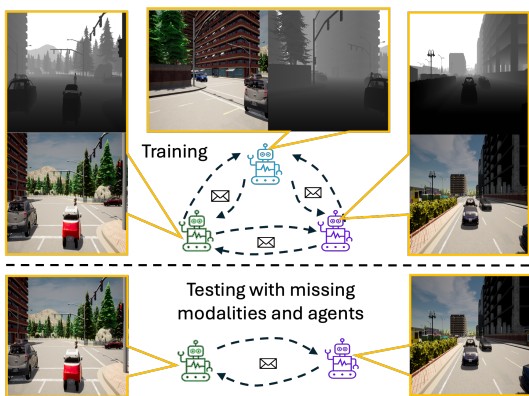

Figure 1: **Illustration of CAML. CAML** enables (1) multiple agents to collaborate and share *multi-modal data during training* while allowing for runtime *inference with reduced modalities during testing*; (2) the *number of agents can vary between training and testing*, ensuring flexibility and robustness in deployment.

tion. The distillation process embeds rich, full-modality knowledge into the student model, preserving the benefits of collaborative training and ensuring reliable inference even when some modalities are unavailable. For example, in autonomous driving, multiple vehicles can share LiDAR and RGB data during training to build robust shared representations, while at deployment each vehicle performs inference using only RGB inputs.

Unlike prior work that either focuses on multi-agent collaboration but without addressing missing modality at test time, or explores AML ideas solely in single-agent settings, **CAML** unifies these concepts. To the best of our knowledge, this work is the first to propose a flexible and principled framework for multi-modal multi-agent systems where limited modalities are available for inference. Overall, this work offers the following key contributions:

- We introduce **CAML**, a novel framework for multi-agent systems that allows agents to share multi-modal data during training, while supporting reduced-modality inference during testing. This collaborative approach enhances data coverage, improves robustness to missing modalities, which is crucial for deployment in in dynamic, resource-constrained environments.

- We validate **CAML** through extensive experiments in collaborative decision-making for connected autonomous driving in accident-prone scenarios, and collaborative semantic segmentation for real-world data of aerial-ground robots. **CAML** achieves up to **58.1**% improvement in *accident detection* for autonomous driving, and up to **10.6**% improvement for more accurate *semantic segmentation*.

## 2 Related Work

**Multi-Agent Collaboration.** Collaboration in multi-agent systems has been widely studied across fields such as autonomous driving and robotics. In autonomous driving, prior research has explored various strategies, including spatio-temporal graph neural networks [8], LiDAR-based end-to-end systems [5], decentralized cooperative lane-changing [35] and game-theoretic models [13]. In robotics, Mandi et al. [34] presented a hierarchical multi-robot collaboration approach using large language models, while Zhou et al. [49] proposed a perception framework for multi-robot systems built on graph neural networks. A review of multi-robot systems in search and rescue operations

was provided by [39], and Bae et al. [1] developed a reinforcement learning (RL) method for multi-robot path planning. Additionally, various communication mechanisms, such as Who2com [31], When2com [30], and Where2comm [17], have been created to optimize agent interactions.

Despite these advancements, existing multi-agent collaboration frameworks remain limited by their focus on specific tasks and the assumption that agents will have consistent access to the same data modalities during both training and testing, an assumption that may not hold in real-world applications. To address these gaps, **CAML** enables agents to collaborate during training by sharing multi-modal data, but at test time, each agent performs inference using reduced modality. This reduces the dependency on certain modalities for deployment, while still allowing agents to leverage additional information during training to enhance overall performance and robustness.

**Auxiliary Modality Learning.** Auxiliary Modality Learning (AML) [41] has emerged as an effective solution to reduce computational costs and the amount of input data required for inference. By utilizing auxiliary modalities during training, AML minimizes reliance on those modalities at inference time. For example, Hoffman et al. [16] introduced a method that incorporates depth images during training to enhance test-time RGB-only detection models. Similarly, Wang et al. [45] proposed PM-GANs to learn a full-modal representation using data from partial modalities, while Garcia et al. [9, 10] developed approaches that use depth and RGB videos during training but rely solely on RGB data for testing. Piasco et al. [37] created a localization system that predicts depth maps from RGB query images at test time. Building on these works, Shen et al. [41] formalized the AML framework, systematically classifying auxiliary modality types and AML architectures.

However, existing AML frameworks are typically designed for single-agent settings, failing to exploit the potential benefits of multi-agent collaboration for improving multi-modal learning. Transitioning from single-agent to multi-agent learning is fundamentally non-trivial, as multi-agent systems introduce unique challenges such as dynamic team compositions and team-level decision-making. **CAML** effectively addresses these challenges and allows agents to collaboratively learn richer multi-modal representations. By sharing complementary information, **CAML** enhances overall data coverage and mitigates information loss during reduced-modality inference, which is particularly important when an individual agent's observations are incomplete due to occlusions or limited sensor range.

**Knowledge Distillation.** Knowledge distillation (KD) [15] is a widely used technique in many domains to reduce computation by transferring knowledge from a large, complex model (teacher) to a simpler model (student). In computer vision, Gou et al. [11] provided a comprehensive survey of KD applications, while Beyer et al. [2] conducted an empirical investigation to develop a robust and effective recipe for making large-scale models more practical. Additionally, Tung and Mori [44] introduced a KD loss function that aligns the training of a student network with input pairs producing similar activation in the teacher network. In natural language processing, Xu et al. [47] reviewed the applications of KD in LLMs, while Sun et al. [42] proposed a Patient KD method to compress larger models into lightweight counterparts that maintain effectiveness. Hahn and Choi [12] suggested a self-distillation approach that leverages the soft target probabilities of a model to train other neural networks, while Liu et al. [25] explored KD under uncertainty. In autonomous driving, Lan and Tian [19] presented an approach for visual detection, Cho et al. [4], Sautier et al. [40] used KD for 3D object detection.

Notice that existing KD mostly distills knowledge from a larger model to a smaller one to reduce computation, Shen et al. [41] aimed to design a cross-modal learning approach using KD to utilize the hidden information from auxiliary modalities within the AML framework. But AML is limited by the scope of a single-agent paradigm. In contrast, we leverage KD within multi-agent settings, where the teacher models are trained with access to shared multi-modal data (e.g., RGB and LiDAR) from multiple agents. By distilling this collaborative knowledge into each agent's reduced modality (e.g., RGB), **CAML** enables robust inference during deployment, even with fewer modalities. This collaborative distillation process enhances each agent's performance by providing richer, complementary knowledge from the collaborative training phase.

## 3 Collaborative Auxiliary Modality Learning

In single-agent frameworks such as AML [41], the missing modalities during testing are referred to as auxiliary modalities, while those that remain available are called the main modality. In contrast, in our framework **CAML**, each agent can process a different number of modalities during training and

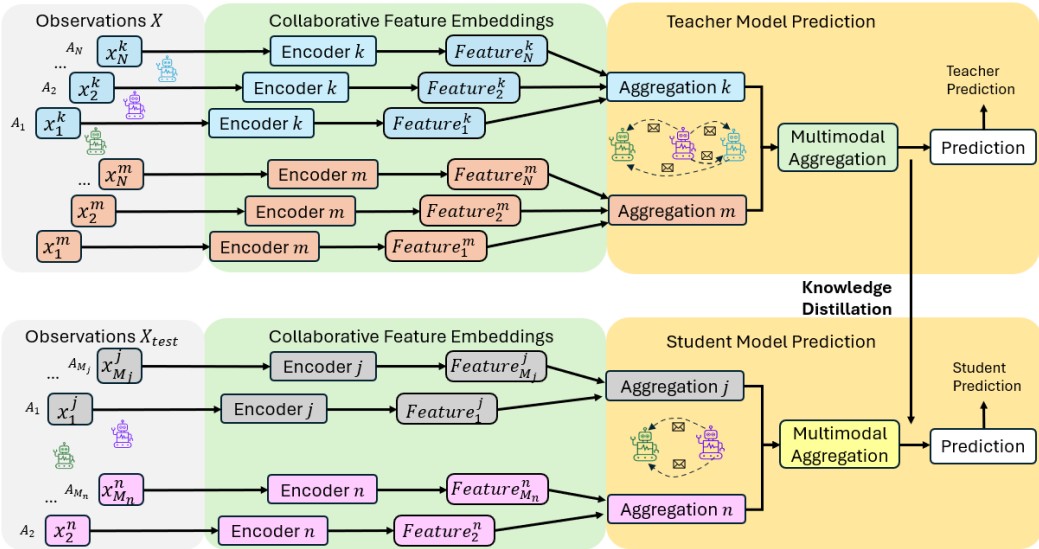

Figure 2: **Overview of the Pipeline of CAML.** The teacher model (top) aggregates and shares multi-modal embeddings across agents to make predictions using the full set of modalities. In contrast, the student model (bottom) processes a subset of modalities per agent and shares them to form a multi-modal embedding. Through knowledge distillation from the teacher, the student learns to produce robust predictions despite missing modalities, enabling effective inference during deployment. In the teacher model, the set of agents is denoted as $\mathcal{A}_{\text{train}} = \{\mathcal{A}_1, \mathcal{A}_2, \dots, \mathcal{A}_N\}$. The set of modalities is denoted as $\mathcal{I}_{\text{train}}$. The observations of all agents are denoted as $X = \{x_1, x_2, \dots, x_N\}$, where $x_i^k$ is the observation acquired by the $i$-th agent $\mathcal{A}_i \in \mathcal{A}_{\text{train}}$ for the $k$-th modality. In the student model, the set of agents is denoted as $\mathcal{A}_{\text{test}} = \{\mathcal{A}_1, \mathcal{A}_2, \dots, \mathcal{A}_M\}$. The set of modalities is denoted as $\mathcal{I}_{\text{test}}$, which is a subset of $\mathcal{I}_{\text{train}}$. The set of agents that have access to the $j$-th modality $\mathcal{I}_j \in \mathcal{I}_{\text{test}}$ is denoted as $\mathcal{A}_{\text{test}}^{\mathcal{I}_j}$, and the number of agents in this set is given by $|\mathcal{A}_{\text{test}}^{\mathcal{I}_j}| = M_j$, with $x_{M_j}^j$ represents the observation acquired by the $M_j$-th agent $\mathcal{A}_{M_j} \in \mathcal{A}_{\text{test}}$ for the $j$-th modality.

different agents can have different main modalities and auxiliary modalities. There is no correlation between the number of agents and the number of modalities.

**Problem Formulation.** We define our problem in both training and testing phases. In the training phase, we consider a multi-agent system with $N$ agents collaboratively completing a task. The set of agents is denoted as $\mathcal{A}_{\text{train}} = \{\mathcal{A}_1, \mathcal{A}_2, \dots, \mathcal{A}_N\}$. The observations of all agents are denoted as $X = \{x_1, x_2, \dots, x_N\}$, where $x_i$ is the observation acquired by the $i$-th agent $\mathcal{A}_i \in \mathcal{A}_{\text{train}}$. The ground truth is denoted as $Y$, which can be an object label, semantic class, or a control command (e.g., brake for an autonomous vehicle). The set of modalities is denoted as $\mathcal{I}_{\text{train}} = \{\mathcal{I}_1, \mathcal{I}_2, \dots, \mathcal{I}_K\}$, such as RGB, LiDAR, Depth, etc, where $K$ is the number of modalities available during training. During training, each agent has access to all these $K$ modalities. In the testing phase, we assume there are $M$ agents. The set of test agents is denoted as $\mathcal{A}_{\text{test}} = \{\mathcal{A}_1, \mathcal{A}_2, \dots, \mathcal{A}_M\}$. In addition, the set of modalities is denoted as $\mathcal{I}_{\text{test}}$, which is a subset of $\mathcal{I}_{\text{train}}$. The number of modalities available during testing is denoted as $L$, where $L \leq K$. The set of agents that have access to the $j$-th modality $\mathcal{I}_j \in \mathcal{I}_{\text{test}}$ is denoted as $\mathcal{A}_{\text{test}}^{\mathcal{I}_j}$, where $\mathcal{A}_{\text{test}}^{\mathcal{I}_j} \in \mathcal{A}_{\text{test}}$, and the number of agents in this set is given by $|\mathcal{A}_{\text{test}}^{\mathcal{I}_j}| = M_j$. This means that during testing, each agent may have access to different number of modalities.

**Approach.** Given the problem definition, we aim to estimate the posterior distribution $P(y|X)$ of the ground truth label $y$ given all agents' observations $X$. During training, we first train a teacher model where each agent has access to all modalities in $\mathcal{I}_{\text{train}}$ and then a student model where each agent has access to partial modalities in $\mathcal{I}_{\text{test}}$. We employ Knowledge Distillation (KD) to transfer the richer multi-modal knowledge from the teacher to the student, enabling the student to benefit from information beyond its own input, as illustrated in Fig. 2. At test time, we perform inference using the student model, which operates solely on the available test modality observations $X_{\text{test}}$.

Specifically, in the teacher model, each agent has access to all multi-modal observations and independently processes its local observations to produce embeddings. These embeddings are then shared among agents based on whether the system operates in a centralized or decentralized manner. If the system is centralized, all collaborative agents share their embeddings with one designated ego agent for centralized processing. If the system is decentralized, each agent shares the embeddings with nearby agents. Subsequently, the shared embeddings corresponding to the same modality are aggregated together. Then we fuse (e.g., via concatenation or cross-attention) the aggregated embeddings of different modalities to create a comprehensive multi-modal embedding. This multi-modal embedding is then passed through a prediction module to produce the teacher model's final prediction. The student model follows a similar network architecture as the teacher. However, instead of processing all modalities, each agent processes only a single or a subset of modalities, which can vary across agents. By sharing these embeddings among agents, the student model also constructs a multi-modal embedding, leveraging the different modalities observed by various agents. This multi-modal embedding is then used to generate the student model's prediction. Our approach enables the student model to maintain effective performance despite missing modalities during testing, significantly enhancing its robustness.

We further provide an intuitive analysis offering insights and explanations to justify the effectiveness of **CAML** by examining how our approach enhances data coverage and leads to greater information gain compared to single-agent approaches. Specifically, we show that aggregating observations from multiple agents enables broader coverage of the data space, capturing complementary information that would be missed by an individual agent. From an information theory perspective, we demonstrate that joint observations from multiple agents generally yield higher mutual information with respect to the ground truth, providing a more informative signal. Please refer to Appendix A.1 for more details.

**System Complexity.** After designing the **CAML** framework, we present a detailed complexity analysis. If the system is centralized, all collaborative agents share their data with one designated ego agent for centralized processing. Each of the $N-1$ collaborative agents performs its local computation independently, with a time complexity of $O(T_c)$ and a space complexity of $O(S_c)$, where $T_c$ represents the time required for local computation, and $S_c$ is the associated space. Thus, the total computation time and space complexities for all collaborative agents are $O(T_c(N-1))$ and $O(S_c(N-1))$, respectively. For simplicity, assuming each communication from one collaborative agent to the ego agent consumes $O(D)$ time complexity and $O(M)$ space complexity, where $D$ is the time required for communication and $M$ is the corresponding space. Therefore, the total communication time and space complexities for gathering information at the ego agent are $O(D(N-1))$ and $O(M(N-1))$, respectively. Then the ego agent aggregates the received data, running a model, having a time and space complexity $O(T_e)$ and $O(S_e)$, where $T_e$ and $S_e$ represent the time and space required for the ego agent's computation. So the total time and space complexities are $O(T_c(N-1)+D(N-1)+T_e)$ and $O(S_c(N-1)+M(N-1)+S_e)$, respectively.

If the system is decentralized, each agent performs its local computation and shares information with nearby agents. For simplicity, let the local computation for a single agent has a time complexity of $O(T)$, where $T$ is the time required for local computation. Assume that communication from one agent to another agent requires $O(D)$ time complexity and $O(M)$ space complexity, where $D$ represents the time of communication between two agents, and $M$ denotes the space required for such communication. For $N$ agents, the total computation time complexity is $O(NT)$. In the worst case, each agent share data with all other agents, this can result in $O(N^2D)$ for pairwise sharing. So the total time complexity is $O(NT + N^2D)$. For space complexity, the storage requirement for all agents is $O(NS)$, where $S$ is the space needed per agent. Communication between agents adds an additional complexity of $O(N^2M)$. So the total space complexity is $O(NS + N^2M)$. In the typical case, if each agent communicates with only other $k$ agents ($k \ll N$) rather than all $N-1$ agents. The total time and space complexities become $O(NT + NkD)$ and $O(NS + NkM)$, respectively.

## 4 Experiments

### 4.1 Collaborative Decision-Making

To evaluate our approach, we first focus on collaborative decision-making in connected autonomous driving (CAV). This involves making critical decisions for the ego vehicle in accident-prone scenarios, such as determining whether or not to take a braking action.

**Data Collection.** The dataset is generated using the AUTOCASTSIM benchmark [38], which features three complex and accident-prone traffic scenarios designed for CAV. These scenarios are characterized by limited sensor coverage and obstructed views, as illustrated in Fig. 6. Following prior works COOPERNAUT [5] and STGN [8], we employ an expert agent with full access to the traffic scene information to collect data. This expert agent is deployed in the environment to generate the data trails, where each trail captures RGB images and LiDAR point clouds from both the ego vehicle and collaborative vehicles, along with the control actions of the ego vehicle. In line with the setup used in STGN [8], the ego vehicle can have up to three collaborative vehicles at each timestep, provided they are within a 150-meter radius. Each trail represents a unique instantiation of the traffic scenario, with differences arising from the randomized scenario configurations. For further dataset details, please refer to Appendix A.2.1.

**Experimental Setup.** After data collection, we train the models using Behavior Cloning (BC) [3, 23], following the setup established by prior works COOPERNAUT [5] and STGN [8]. For each vehicle, both RGB and LiDAR data are used during training, while only RGB data is used during testing in **CAML**. Please refer to Appendix A.2.3 for more details on the BC procedure. All experiments are conducted across four repeated runs, and we report the mean and standard deviation of the results.

For processing RGB data, we first resize the image to $224 \times 224$ and use ResNet-18 [14] as the encoder to extract a feature map for each vehicle. We then apply self-attention on the feature map to dynamically compute the importance of features at different locations. After the self-attention, we apply three convolution layers with each followed by a ReLU activation. Finally, we obtain a 256-d feature representation after passing through a fully connected layer. To aggregate RGB feature embeddings from connected vehicles to the ego vehicle, we use the cross-attention mechanism. For processing the LiDAR data, we use the Point Transformer as the encoder for each vehicle and utilize the COOPERNAUT [5] model to aggregate LiDAR feature embeddings between connected vehicles. Then we concatenate the final RGB and LiDAR embeddings for the ego vehicle's decision-making, with a three-layer MLP as the prediction module to output the action.

For the training of Knowledge Distillation (KD), we first train a teacher model offline using a binary cross-entropy loss, where each vehicle has both RGB and LiDAR data. Then we train a student model to mimic the behavior of the teacher model with only RGB data for each vehicle. For each data sample, the student model receives the same RGB image that the teacher model was given. For further details on the KD training process, please refer to Appendix A.5. And for the detailed training settings, please see Appendix A.4. We employ the following two metrics for evaluation: (1) **Accident Detection Rate (ADR)**: This is the ratio of accident-prone cases correctly detected by the model compared to the total ground truth accident-prone cases. An accident-prone case is identified when the ego vehicle performs a braking action. This metric measures the model's effectiveness in identifying potential accidents. (2) **Expert Imitation Rate (EIR)**: This denotes the percentage of actions accurately replicated by the model out of the total expert actions. It serves to evaluate how well the model mimics expert driving behavior.

**Baselines.** We implement the following baselines for comparison: (1) **AML** [41]: In the AML setting, the ego vehicle operates independently without collaboration with other vehicles (non-collaborative). Both RGB and LiDAR data are available during training for the vehicle, while only RGB data is available during testing. (2) **COOPERNAUT** [5]: A collaborative method that processes LiDAR data during both training and testing. It employs the Point Transformer [48] as the backbone, encoding raw 3D point clouds into keypoints. (3) **STGN** [8]: A collaborative approach that utilizes spatial-temporal graph networks for decision-making, with RGBD data used for both training and testing.

**Baselines Comparison.** How well does **CAML** perform against other methods for decision-making in CAV? We evaluate **CAML** against the baselines and present the results in Fig. 3, which demonstrate a clear performance advantage of **CAML** across all three accident-prone scenarios.

Compared to single-agent systems like AML, **CAML** achieves notable improvements in mean ADR: **13.0**% in the overtaking scenario, **34.6**% in the left turn scenario, and a significant **58.1**% in the red light violation scenario. These improvements highlight the advantages of **CAML**'s collaborative framework, which enables the ego vehicle to aggregate complementary sensory data from connected vehicles. This richer, multi-agent perspective significantly enhances situational awareness, allowing the ego vehicle to detect potential accidents and respond proactively, such as

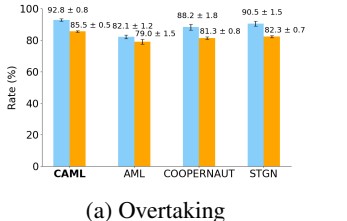
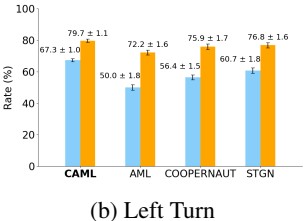
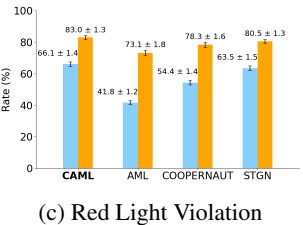

| (a) Overtaking | (b) Left Turn | (c) Red Light Violation |

Accident Detection Rate (ADR)  ▉  Expert Imitation Rate (EIR)

Figure 3: **Performance Comparison of CAML Against Baselines.** We evaluate performance using two metrics: Accident Detection Rate (ADR) and Expert Imitation Rate (EIR) across three accident-prone scenarios: (a) Overtaking, (b) Left Turn, and (c) Red Light Violation. *CAML demonstrates superior performance across all scenarios compared to these baselines by up to $58.1\%$, benefiting considerably from the multi-modal multi-agent collaboration.*

braking in time to avoid collisions, especially in scenarios involving occlusions or restricted views. When compared to COOPERNAUT, **CAML** achieves improvements in mean ADR by up to $21.5\%$, further demonstrating the strength of its collaborative and multi-modal approach.

Additionally, how does **CAML** compare with other approaches that have access to more modalities during testing? We evaluate **CAML** against STGN [8] under multi-agent settings. **CAML** uses only RGB data during testing but STGN uses both RGB and depth data. Despite with fewer modality, **CAML** achieves comparable, and in some cases superior performance while relying solely on RGB data, as illustrated in Fig. 4. Notably, **CAML** exceeds the mean ADR of STGN by $10.9\%$ in the left-turn scenario, demonstrating that our model can enhance driving safety even when constrained to fewer modalities. This further underscores the strength of **CAML**, which effectively leverages LiDAR data as an auxiliary modality during training to boost performance. The fact that **CAML** matches or exceeds the performance of a model that uses more data at test time highlights the modality efficacy of our approach.

**System Generalizability.** How effectively does the system generalize when we have fewer agents during testing compared to training? (e.g., we have multi-agent collaboration during training but only single agent during testing). We test the case where multi-agent collaboration is used during training, but only a single agent is present during testing. This test is also motivated by practical constraints, where in many real-world situations, multi-vehicle connected systems are not available, we only have a single vehicle. But it is reasonable to have multi-vehicle connected systems with multiple modalities during training to develop robust models. After training, we can then apply the model on a single vehicle for testing, which is very valuable in practice and provides a cost-effective solution.

We compare the performance of our approach with COOPERNAUT [5] and STGN [8] under single-agent settings, with results shown in Fig. 4. **CAML** outperforms the other baselines across all scenarios, for both ADR and EIR metrics. This demonstrates that even with a single agent during testing, **CAML** remains highly effective, by utilizing the multi-agent collaboration and auxiliary modalities provided by the teacher model during training.

**Efficiency Analysis.** We evaluate the communication and inference efficiency of **CAML** compared to other cooperative approaches, COOPERNAUT [5] and STGN [8]. To assess communication bandwidth, we follow STGN [8] and use the shared package size (PS) metric, which measures the size of data exchanged between connected vehicles, serving as an indicator of communication efficiency

Table 1: **Communication and inference efficiency comparison.** **CAML** can operate more efficiently with *lower communication overhead and faster inference than both* [5] *and* [8].

| Approach | PS (KB) | Latency (ms) |
|---|---|---|
| COOPERNAUT [5] | 65.5 | 90.0 |
| STGN [8] | 4.9 | 18.5 |
| **CAML** | 1.0 | 3.7 |

in collaborative decision-making. For inference efficiency, we measure latency per batch, which reflects the average time required to process a batch during inference. The evaluation results are summarized in Table 1. As shown, **CAML** achieves lower PS and latency compared to other approaches, highlighting its superior communication and inference efficiency.

**Ablation Studies.** In the ablation studies, we first compare the performance of **CAML** with two other baselines: the fully-equipped teacher model, and a multi-agent BC model trained and

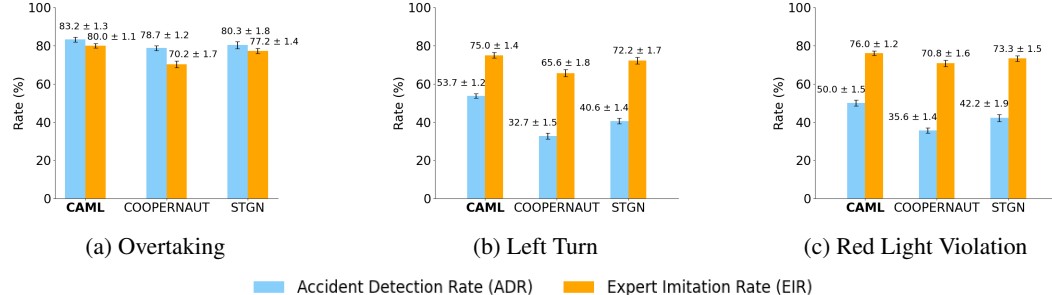

Figure 4: **Single-Agent System Generalizability of CAML.** We evaluate the generalizability of **CAML** by testing the case where we have multi-agent collaboration during training, but only a single agent during testing. We compare the performance of our approach with COOPERNAUT and STGN under single-agent settings. *CAML with a single agent during testing consistently outperforms the other baselines across all scenarios, offering a valuable and cost-effective solution for practical applications.*

tested using only RGB images, without knowledge distillation (KD). As shown in Table 2, while **CAML** shows a slight performance drop compared to the teacher, due to missing modalities at inference, it still achieves strong and robust performance. This highlights the effectiveness of our method in handling modality reduction scenarios, making it especially suitable for resource-constrained environments. Furthermore, **CAML** consistently outperforms the RGB-only BC baseline. The results demonstrate the advantage of incorporating multi-modal learning into multi-agent systems. Specifically, leveraging additional modalities during training through knowledge distillation allows **CAML** to encode richer information into the student model, resulting in improved performance at test time.

Next, we evaluate the case involving more modalities beyond RGB and LiDAR. Each vehicle now receives RGB, LiDAR, and state information (e.g., position and velocity) during training, while only RGB inputs are available during inference. As shown in Table 3, incorporating state information as an additional modality further improves system performance, demonstrating **CAML**'s ability to effectively leverage richer multimodal data during training while maintaining strong performance under reduced-modality conditions.

Table 2: Performance comparison (%) of **CAML** with the teacher model and the RGB-only BC baseline. While **CAML** shows a slight performance drop compared to the teacher model due to missing modalities at inference, *it still achieves strong and robust performance. Moreover, CAML outperforms the RGB-only BC baseline across all scenarios.*

| Approach | Overtaking | | Left Turn | | Red Light Violation | |
|---|---|---|---|---|---|---|
| | ADR↑ | IR↑ | ADR↑ | IR↑ | ADR↑ | IR↑ |
| Teacher | 96.0±0.3 | 86.8±0.4 | 73.3±0.8 | 81.2±0.6 | 68.8±0.9 | 85.8±0.7 |
| RGB-only BC | 85.2±1.0 | 83.2±0.9 | 56.4±1.5 | 78.0±1.2 | 55.8±1.3 | 81.0±1.2 |
| **CAML** | 92.8±0.8 | 85.5±0.5 | 67.3±1.0 | 79.7±1.1 | 66.1±1.4 | 83.0±1.3 |

We also investigate the complementary role of different modalities by retaining LiDAR and removing RGB during testing. As presented in Table 4, the performance of retaining LiDAR is slightly better than retaining RGB, as LiDAR provides more precise spatial localization, which is beneficial for accident detection.

Table 3: Performance evaluation with three modalities: RGB, LiDAR, and state information. Incorporating state information as an additional modality further enhances overall system performance.

| Modalities | Overtaking | | Left Turn | | Red Light Violation | |
|---|---|---|---|---|---|---|
| | ADR↑ | IR↑ | ADR↑ | IR↑ | ADR↑ | IR↑ |
| RGB+LiDAR | 92.8±0.8 | 85.5±0.5 | 67.3±1.0 | 79.7±1.1 | 66.1±1.4 | 83.0±1.3 |
| RGB+LiDAR+State | 94.0±0.7 | 86.4±0.6 | 69.7±1.2 | 81.7±1.3 | 68.3±1.2 | 84.7±1.0 |

Furthermore, we compare **CAML**, which adopts an intermediate cooperation strategy, with a late cooperation baseline. In the late cooperation setting, each vehicle independently processes RGB and LiDAR inputs using the same encoders as in **CAML** (ResNet-18 for RGB and PointTransformer for LiDAR), followed by MLP heads to generate action logits. For each modality, we first fuse the logits from connected vehicles via

Table 4: Performance comparison (%) of **CAML** when retaining either RGB or LiDAR during testing. Both settings achieve comparable performance, but retaining LiDAR yields slightly better results due to its more precise spatial localization, which benefits accident detection.

| Modality | Overtaking | | Left Turn | | Red Light Violation | |
|---|---|---|---|---|---|---|
| | ADR↑ | IR↑ | ADR↑ | IR↑ | ADR↑ | IR↑ |
| RGB | 92.8±0.8 | 85.5±0.5 | 67.3±1.0 | 79.7±1.1 | 66.1±1.4 | 83.0±1.3 |
| LiDAR | 93.3±0.9 | 86.1±0.6 | 68.0±0.9 | 80.4±1.3 | 67.2±1.2 | 83.8±1.4 |

averaging and then fuse the averaged logits across modalities to determine the ego vehicle's control actions. We apply the same KD procedure as in **CAML**, distilling from the RGB–LiDAR teacher model to a student model that uses only RGB at test time. As shown in Table 5, **CAML** outperforms

the late cooperation approach by enabling the model to learn richer and more complementary representations across agents and modalities, whereas late cooperation relies solely on output-level fusion, leading to potential information loss.

Overall, the experimental results clearly illustrate the superiority of our **CAML** framework. The ability of **CAML** to learn a more effective driving policy stems from the collaborative multi-modal behavior of multiple agents, which together capture a wider and more nuanced representation of data. This broader data coverage enables the ego vehicle to make better-informed

Table 5: Performance comparison (%) between **CAML** and the late cooperation approach. **CAML** achieves superior results by enabling the model to learn richer and more complementary representations across agents and modalities.

| Approach | Overtaking | | Left Turn | | Red Light Violation | |
|---|---|---|---|---|---|---|
| | ADR↑ | IR↑ | ADR↑ | IR↑ | ADR↑ | IR↑ |
| **CAML** | 92.8±0.8 | 85.5±0.5 | 67.3±1.0 | 79.7±1.1 | 66.1±1.4 | 83.0±1.3 |
| Late Cooperation | 88.0±1.0 | 83.5±1.0 | 60.2±1.3 | 78.5±0.9 | 58.2±1.5 | 81.7±1.5 |

decisions, improving safety and performance, particularly in complex, dynamic, and accident-prone environments where isolated agents with limited sensing.

### 4.2 Collaborative Semantic Segmentation

To further evaluate our approach, we focus on collaborative semantic segmentation by conducting experiments with real-world data from aerial-ground robots. We use the dataset CoPeD [50], with one aerial robot and one ground robot, in two different real-world scenarios of the indoor NYUARPL and the outdoor HOUSEA. For more details about the dataset, please refer to [50]. Additionally, we introduce noise to the RGBD data collected by the ground robot. For both aerial and ground robots, RGB and depth data are used during training, while only RGB data is used during testing in **CAML**.

**Experimental Setup.** We adopt the FCN [32] architecture as the backbone for semantic segmentation. To process RGB and depth data locally for each robot, we use ResNet-18 [14] as the encoder to extract feature maps of size $7 \times 7$. Please refer to Appendix A.3.1 for more details of the experimental setup. We first train a teacher model offline with aerial-ground robots collaboration using cross-entropy loss, where each robot has both RGB and depth data. Then we train a student model to mimic the behavior of the teacher model with only RGB data for both aerial and ground robots through KD. The KD process is similar to that of the collaborative decision-making in CAV, but here we use a cross-entropy loss as the student task loss. Please see Appendix A.4 for the detailed training settings. We evaluate performance using the **Mean Intersection over Union (mIoU)** metric, which quantifies the average overlap between predicted segmentation outputs and ground truth across all classes. We compare the performance of **CAML** with other baselines including AML [41] and FCN [32]. In the AML approach, only the ground robot operates, with RGB and depth data available during training but only RGB data used for testing. The FCN approach involves only the ground robot operating with RGB data for both training and testing.

**Experimental Results.** We first present the experimental results of baselines comparison in Table 6, where **CAML** demonstrates superior performance in terms of mIoU across both indoor and outdoor environments. Specifically, **CAML** achieves an improvement of mIoU for $8.9\%$ in indoor scenario and $10.6\%$ in outdoor scenario compared to AML [41]. We also show the qualitative results in Fig. 5. As we can see, despite the noisy input image from the ground robot, **CAML** produces predictions that are closest to the ground truth. This improvement is

Table 6: **Baseline Comparison of Semantic Segmentation** on real-world dataset CoPeD [50] using aerial-ground robots in indoor and outdoor environments. *CAML achieves the highest mIoU in both environments, with upto* **10.6%** *higher accuracy.*

| Approach | mIoU (%) | |
|---|---|---|
| | Indoor | Outdoor |
| FCN [32] | 51.2 | 56.2 |
| AML [41] | 55.9 | 60.3 |
| **CAML** | **60.1** | **66.8** |
| Improvement over SOTA | 4.2-8.9 | 6.5-10.6 |

attributed to **CAML**'s multi-agent collaboration, which provides complementary information to enhance data coverage and offers a more comprehensive understanding of the scenes. Additionally, the utilization of auxiliary depth data during training results in more precise segmentation outputs.

**Ablation Studies.** We also investigate another variant of **CAML**, called Pre-fusion **CAML**, as ablation studies. In this variant, each robot first locally extracts feature maps of size $7 \times 7$ for both RGB and depth modalities. Instead of separately fusing the RGB and depth features between the robots, we first fuse the feature maps of RGB and depth within each single robot using cross-attention. Then we share and merge the fused RGBD features between robots via concatenation. We also apply $1 \times 1$ convolution to reduce the feature maps to the original channel dimensions. The multi-modal multi-agent feature aggregations then pass through the decoder. Finally, we obtain the output map by

| Input | Ground truth | FCN | AML | CAML |
|-------|--------------|-----|-----|------|

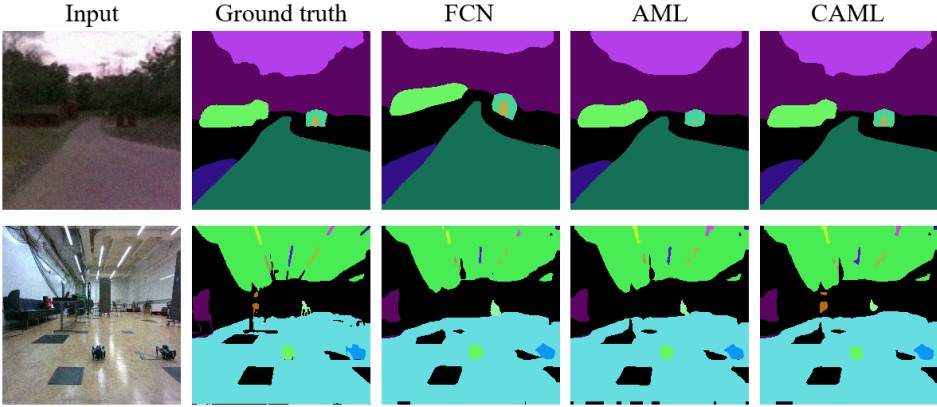

Figure 5: Qualitative results of different approaches on semantic segmentation on real-world data from aerial-ground robots in scenarios of both indoor and outdoor environments. From left to right, input image for the ground robot, ground truth segmentation map, FCN prediction, AML prediction, and **CAML** prediction. *CAML prediction is the closest to the ground truth.*

upsampling to match the input image size. The mIoU of the Pre-fusion **CAML** is similar to that of **CAML**, achieving $59.2\%$ and $65.8\%$ for indoor and outdoor environments, respectively. Although the fusion order is different, both versions benefit from robust feature aggregation and multi-agent collaboration, which ultimately results in better segmentation performance. And **CAML** can easily shift to Pre-fusion **CAML** because of the flexibility of our framework. Please refer to Appendix A.3.2 for more details.

## 5 Conclusions

In conclusion, we propose Collaborative Auxiliary Modality Learning (**CAML**), a unified framework for multi-modal multi-agent systems. Unlike prior methods that either focus on multi-agent collaboration without modality reduction or address multi-modal learning in single-agent settings, **CAML** integrates both aspects. It enables agents to collaborate using shared modalities during training while allowing efficient, modality-reduced inference. This not only lowers computational costs and data requirements at test time but also enhances predictive accuracy through multi-agent collaboration. We provide an intuitive analysis of **CAML** in terms of data coverage and information gain, justifying its effectiveness. **CAML** demonstrates up to a $58.1\%$ improvement in accident detection for connected autonomous driving in complex scenarios and up to a $10.6\%$ mIoU gain in real-world aerial-ground collaborative semantic segmentation. These improvements underscore the practical implications of our framework, especially for resource-constrained environments. For a discussion on limitations and future work, please see Appendix A.6.

## Acknowledgment

This research is supported in part by National Science Foundation, Dr. Barry Mersky & Capital One E-Nnovate Endowed Professorships.

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

# A Appendix

## A.1 Analysis

We provide an intuitive analysis to justify the effectiveness of **CAML** and illustrate why it outperforms single-agent approaches such as AML. Please note that this analysis is intended to offer intuitive insights and explanations into the benefits of **CAML**, rather than serving as a formal mathematical proof, as this may be intractable in this context.

We analyze from two key perspectives: **Data Coverage** and **Information Gain**. We aim to address the following key questions: (a) Does the collaboration of multiple agents provide complementary information that increases data coverage? Specifically, does combining observations from each agent lead to a more accurate and comprehensive prediction compared to using a single agent? (b) Does the collaboration increase the mutual information between the observations and the true label?

**Data Coverage.** To address Question (a), we study data coverage and information provided by each agent in a multi-agent system. Let the entire data space be denoted as $\mathcal{D}$, which consists of various subsets. Each agent $\mathcal{A}_i$ in the system covers a subset of this data space: $\mathcal{C}_i \subseteq \mathcal{D}$. The overall coverage by the system is given by the union of all subsets covered by individual agents: $\mathcal{C}_{multi} = \cup_{i=1}^{N} \mathcal{C}_i$. This ensures that $|\mathcal{C}_{multi}| \geq \max |\mathcal{C}_i|$. If only a single agent is available, it can only observe a portion of the data space, leaving parts of the space unobserved, which leads to incomplete information for estimating the true label $y$. We show a qualitative example of multi-agent collaboration providing complementary information to enhance data coverage in Fig. 7 in Appendix A.2.2.

From a probabilistic perspective, when multi-agent collaboration is in place, the combined likelihood $P(X|y)$ is modeled as a multivariate distribution. This approach provides a broader and more accurate representation of the data space by integrating information from all agents and modeling the dependencies and correlations between them. Compared to a univariate distribution $P(x_i|y)$ for a single agent $\mathcal{A}_i$, the multivariate distribution covers a larger portion of the data space $\mathcal{D}$, thus enhancing data coverage. This allows the exploration of more complex patterns, relationships, and complementary information from different agents. By capturing a richer set of interactions and correlations among the agents' observations, the multivariate distribution supports more informed decision-making. The model's predictions are based on a comprehensive view of the environment, thus leading to more accurate outcomes.

**Information Gain.** To address Question (b) about information gain, we analyze using information theory. Let $I(y; x_i)$ represent the mutual information between the true label $y$ and agent $\mathcal{A}_i$'s observation $x_i$, which quantifies how much information $x_i$ provides about the estimation of $y$. The joint mutual information between $y$ and the set of all observations $X$ is $I(y; X)$. In the context of multi-agent collaboration, the joint observations $X$ from multiple agents typically provide more comprehensive information about the true label $y$ compared to the observation of any single agent. Therefore, the mutual information $I(y; X)$ is always greater than or equal to the mutual information from a single agent: $I(y; X) \geq I(y; x_i)$. We can formally justify this using the chain rule for mutual information:

$$
\begin{aligned}
I(y; X) &= I(y; x_1, \ldots, x_n) \\
&= I(y; x_1) + I(y; x_2 \mid x_1) + \cdots + I(y; x_n \mid x_1, \ldots, x_{n-1}).
\end{aligned}
\tag{1}
$$

Each term ($I(y; x_k \mid x_1, \ldots, x_{k-1}) \geq 0$) is non-negative because mutual information is always non-negative. It follows that:

$$
I(y; X) \geq I(y; x_i) \quad \text{for any individual } x_i.
\tag{2}
$$

Adding more observations (e.g., expanding from $x_i$ to $X$) does not reduce the information about $y$. Even if some observations are redundant (e.g., $x_j$ duplicates $x_i$), the joint mutual information $I(y; X)$ remains at least as large as $I(y; x_i)$. Thus, the combined observations from multi-agent collaboration generally provide more information about $y$ than a single observation, improving the overall estimate.

### A.2 Connected Autonomous Driving

#### A.2.1 Dataset Details

The dataset is generated using the AUTOCASTSIM [38] benchmark, which features three complex and accident-prone traffic scenarios for connected autonomous driving, characterized by limited sensor coverage or obstructed views. These scenarios are realistic and include background traffic of 30 vehicles. They involve challenging interactions such as overtaking, lane changing, and red-light violations, which inherently increase the risk of accidents: (1) **Overtaking**: A sedan is blocked by a truck on a narrow, two-way road with a dashed centerline. The truck also obscures the sedan's view of oncoming traffic. The ego vehicle must decide when and how to safely pass the truck. (2) **Left Turn**: The ego vehicle attempts a left turn at a yield sign. Its view is partially blocked by a truck waiting in the opposite left-turn lane, reducing visibility of vehicles coming from the opposite direction. (3) **Red Light Violation**: As the ego vehicle crosses an intersection, another vehicle runs a red light. Due to nearby vehicles waiting to turn left, the ego vehicle's sensors are unable to detect the violator.

Following the setup established by prior works COOPERNAUT [5] and STGN [8], we collect 24 data trails for each scenario, using 12 trails for training and the remaining 12 for testing. Each trail represents a unique instantiation of the traffic scenario, with differences arising from the randomized scenario configurations. These variations include different types of vehicles present in the scene, varying initial positions and trajectories of collaborative vehicles, difference in traffic flow and interactions, resulting in different visual, LiDAR inputs and control actions for each trail. Each trail contains RGB images and LiDAR point clouds from multiple vehicles, resulting in a substantially large dataset with a total size of approximately 400 GB.

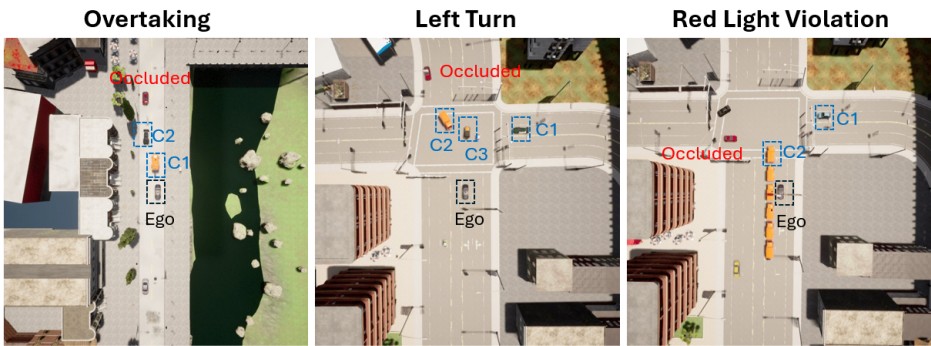

Figure 6: Three accident-prone scenarios in connected autonomous driving benchmark AUTOCAST-SIM: overtaking, left turn, and red light violation.

#### A.2.2 Data Coverage

We present a qualitative example highlighting how multi-agent collaboration provides complementary information to enhance data coverage in Fig. 7. In a red-light violation scenario for connected autonomous driving, as shown in the following figure, the ego vehicle's view is obstructed, rendering the occluded vehicle invisible. However, collaborative vehicles are able to detect the occluded vehicle, providing critical complementary information. This additional data helps the ego vehicle overcome its occluded view, enabling it to make more informed decisions and avoid potential collisions with the occluded vehicle.

#### A.2.3 Behavior Cloning Details

In the CAV task, we apply Behavior Cloning (BC) following prior works COOPERNAUT [5] and STGN [8]. The process involves first running an expert agent to collect data from both the ego vehicle and collaborative vehicles. This data includes RGB images, LiDAR point clouds, and the control actions of the ego vehicle, specifically, braking decisions. Please see more details of the data collection in Appendix A.2.1.

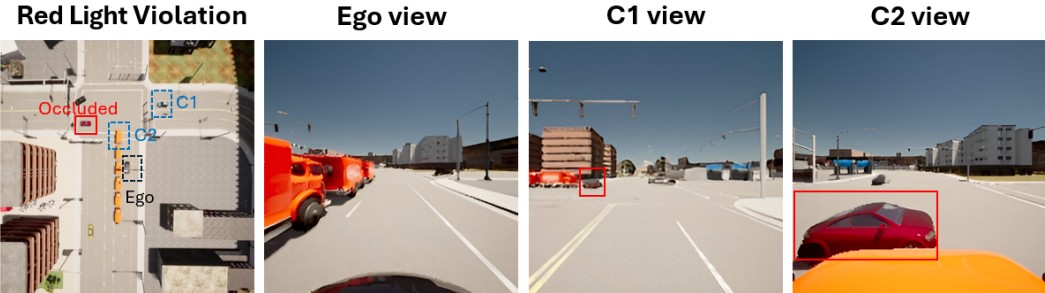

Figure 7: Qualitative example of multi-agent collaboration provides complementary information to enhance data coverage.

To integrate BC into our multi-agent setup, we train a model that takes as input the RGB and LiDAR data from both the ego and collaborative vehicles, and it outputs the control actions for the ego vehicle. BC is trained using a cross-entropy loss to detect accident-prone cases. The purpose of BC is to let the ego vehicle effectively mimic the expert's behavior, allowing it to better assess whether a situation is dangerous or not and make informed decisions, such as braking or continuing to drive. This enhances safety in accident-prone environments by ensuring more accurate and context-aware decision-making.

### A.3 Real-World Aerial-Ground Scenarios

#### A.3.1 Experimental Setup

We resize the input RGB and depth images to $224 \times 224$. To process RGB and depth data locally for each robot, we use ResNet-18 [14] as the encoder to extract feature maps of size $7 \times 7$. The RGB features from both robots are shared and fused through channel-wise concatenation, and the depth features are processed similarly. Then we apply $1 \times 1$ convolution to reduce the fused feature maps to the original channel dimensions for RGB and depth, respectively. We subsequently apply cross-attention to fuse the RGB and depth feature maps to generate multi-agent multi-modal feature aggregations. These aggregated features are passed through the decoder and upsampled to produce an output map matching the input image size.

#### A.3.2 Advantages of CAML and Pre-fusion CAML

Both **CAML** and its variant Pre-fusion **CAML** have their advantages, **CAML** fuses the same modalities across different agents, which provides better alignment because it ensures consistency in feature representation. And this approach is particularly beneficial when individual agent views are limited, as **CAML** effectively leverages diverse viewpoints to provide complementary information, enhancing overall data coverage. On the other hand, Pre-fusion **CAML** allows agent-specific contextual understanding by fusing different modalities locally within each agent. Furthermore, the system avoids redundant communication between agents by transmitting multi-modal aggregated features rather than modality-specific features separately. **CAML** can easily shift to Pre-fusion **CAML** because of the flexibility of our framework, depending on application scenarios.

#### A.4 Training Complexity

We report the training complexity of AML [41] and **CAML** for the experiments of collaborative decision-making in CAV, and collaborative semantic segmentation for aerial-ground robots in Table 7 and Table 8, respectively. For the experiments, we employ a batch size of 32 and the Adam optimizer [18] with an initial learning rate of $1e{-}3$, and a Cosine Annealing Scheduler [33] to adjust the learning rate over time. The model is trained on an Nvidia RTX 3090 GPU with AMD Ryzen 9 5900 CPU and 32 GB RAM for 200 epochs.

Table 7: Training complexity of AML and **CAML** in collaborative decision-making for connected autonomous driving.

| Approach | Parameters | Time/epoch |
|----------|-----------|------------|
| AML [41] | 19.5M | 34s |
| **CAML** | 39.3M | 73s |

Table 8: Training complexity of AML and **CAML** in collaborative semantic segmentation for aerial-ground robots.

| Approach | Parameters | Time/epoch |
|----------|-----------|------------|
| AML [41] | 13.5M | 3s |
| **CAML** | 25.5M | 7s |

## A.5   Knowledge Distillation

We begin by training a teacher decision-making model $\mathcal{T}$ offline using both RGB and LiDAR data, with a binary cross-entropy loss: $\mathcal{L}_{BCE}(y, \mathcal{T}) = -\mathbb{E}_{\mathcal{D}}\big[y_i \log(p_i) + (1 - y_i)\log(1 - p_i)\big]$, where $\mathcal{D}$ is the dataset, $y_i$ is the ground truth indicating whether the vehicle should brake, $p_i$ is the predicted probability by the teacher model $\mathcal{T}$. The student model $\mathcal{S}$ is trained to mimic the behavior of the teacher model while having less modalities. For each data point, the student model receives the same RGB image that the teacher model was given. The loss for the student model is a combination of two terms: the distillation loss using KL divergence between the student output and teacher output (soft targets), and the student task loss, which is the binary cross entropy loss between the student output and the true labels (hard targets). The soft targets from the teacher enrich learning with class similarities, while hard targets ensure alignment with true labels. The soft targets are generated by applying a temperature scaling to the logits. Following prior works [15, 41, 43], the scaled logits are defined as: $z_i = \frac{\exp(z_i/t)}{\exp(z_0/t) + \exp(z_1/t)}$, where $z_i$ is the logit for class $i$ and $t = 4.0$ is the temperature. The distillation loss is defined as: $\mathcal{L}_{KD}(\mathcal{S}, \mathcal{T}) = -\sum z_i^{\mathcal{T}} \log(z_i^{\mathcal{S}})$, where $z_i^{\mathcal{T}}$ and $z_i^{\mathcal{S}}$ are the soft target probability from the teacher and student model, respectively. The overall loss for the student model is a weighted sum of the distillation loss and the binary cross-entropy loss: $\mathcal{L}_{\mathcal{S}} = (1 - \alpha)\mathcal{L}_{BCE}(y, \mathcal{S}) + \alpha t^2 \mathcal{L}_{KD}(\mathcal{S}, \mathcal{T})$ with $\alpha = 0.9$ as the weight. After the training of knowledge distillation process, we obtain a student model that uses only RGB data while learning from a teacher model that has access to both RGB and LiDAR data. This enables the student model to be effective during testing with only RGB data. Additionally, by leveraging knowledge distillation, the student model benefits from the additional insights provided by the LiDAR data during training, learning more effectively compared to training solely with RGB data.

Training a teacher model and then distilling it into a student model allows the student model benefit from the richer information that the teacher model has. Since student models are typically smaller or operate with fewer modalities, they are designed to be more efficient, with fewer parameters or reduced complexity. Deploying such a student model in a resource-constrained environment is crucial, especially when the teacher model is too costly to deploy due to its computation requirements.

## A.6   Limitations and Future Work

**CAML** is capable of handling different and mixed modalities across agents at test time. Feature embeddings are first extracted using modality-specific encoders and then fused across agents (e.g., via concatenation or cross-attention) for downstream decision-making. In the student model of **CAML**, the modality configuration can be specified individually for each agent, allowing flexibility without requiring all agents to share the same modalities. Although the current student model does not dynamically vary modalities over time, such capability can be achieved without modifying the architecture. A promising direction is to incorporate a modality dropout strategy during knowledge distillation, where certain modalities are randomly masked or removed for each agent during training. This would enable the student model to better adapt to dynamically changing modality availability. Moreover, **CAML** can be extended with uncertainty estimation [22, 24] to assess the reliability of each modality and enhance robustness when sensor inputs are noisy or degraded (e.g., under adverse weather conditions in connected autonomous driving).

