# OpenReview forum: "CAML: Collaborative Auxiliary Modality Learning for Multi-Agent Systems"
_NeurIPS.cc/2025/Conference — NeurIPS 2025 poster_

### Official Review · Reviewer_fUqC · 2025-06-01

**Clarity:** 2
**Significance:** 2
**Originality:** 2
**Rating:** 4
**Confidence:** 4

**Summary:**

In this paper, the authors introduce Collaborative Auxiliary Modality Learning (CAML), a framework that allows agents to collaborate and share reduced modalities during inference. The proposed framework is trained using a knowledge distillation approach. The experimental results on collaborative decision-making and semantic segmentation tasks demonstrate the effectiveness of CAML.

**Questions:**

Is CAML an intermediate cooperation approach? If so, what are the key differences between it and other RGB-based intermediate cooperation methods? It appears that the primary distinction lies in the enhancement of RGB representation through knowledge distillation.

Regarding the training complexity of pre-fusion CAML, how does it compare to other methods? Is it a more favorable option that achieves a balance between training complexity and inference performance?

Furthermore, how does CAML's performance compare to that of late cooperation (result-level cooperation), which is also robust to modality missing?

**Ethical Concerns:**

["NO or VERY MINOR ethics concerns only"]

**Final Justification:**

The experimental results address my concerns to a large extent. I have carefully read the authors' response as well as the discussion with other reviewers. I have decided to change the score to relative positive.

**Limitations:**

yes

**Quality:**

2

**Strengths And Weaknesses:**

The proposed CAML extends Auxiliary Modality Learning (AML) to multi-agent cooperative scenarios, enhancing the robustness of multi-agent systems to missing modalities. The proposed approach is technically sound. The performance improvements highlight the effectiveness of CAML.

However, CAML is essentially a simple combination of AML and a cooperative system, and its additional contributions are somewhat limited. The agents in the framework are restricted to the RGB modality, and dynamic modality dropout is not yet supported. This limitation prevents CAML from effectively addressing the challenges posed by dynamic environments (claimed by the authors), such as those encountered in Connected and Autonomous Vehicles (CAV). Technically, CAML can be seen as a cooperative framework with an enhanced RGB representation. The distinctions between CAML and general cooperative paradigms should be clearly articulated. Moreover, a more comprehensive comparison with existing cooperative approaches is needed to better demonstrate its performance advantages.

---

> ### Author Rebuttal · Authors · 2025-07-30
>
> Thanks for your comments and feedback. We are glad that you found our paper **technically sound** with **effective performance improvements**. We address your concerns below and will incorporate all feedback into the final version. We hope you could consider raising the score.
>
> >Contributions of CAML
>
> Please note that CAML is not simple combination of AML and cooperative system. Multi-agent settings introduces substantial complexity, requiring careful consideration of new challenges. CAML specifically addresses the unique challenges of multi-agent systems, including heterogeneity in sensing modalities, dynamic team compositions, and team-level decision-making. CAML has broader applications of connected autonomous driving, collaborative search and rescue, as well as intelligent transportation. To the best of our knowledge, this work is the first to propose a flexible and principled framework capable of tackling a broad range of collaborative tasks in multi-agent systems where limited modalities are available for inference. Unlike prior work that either focuses on multi-agent collaboration but without addressing missing modality at test time, or explores AML ideas solely in single-agent settings, CAML unifies these directions for the first time.
>
> The agents in the framework are not restricted to the RGB modality. CAML is a general framework and can work with any modality. In the experiments of CAV and aerial-ground robot collaboration, we use RGB as the available modality during inference because it is the most commonly used and widely available in robotics and autonomous driving.
>
> Additionally, we conduct new experiments where LiDAR is retained while RGB is removed during testing, please see the results in the following table, with mean and std over four repeated runs. The performances of retaining RGB or LiDAR are similar but retaining LiDAR shows slightly better performance as LiDAR provides more precise spatial localization which is better for accident detection.
>
> |Modality during testing| Overtaking|| Left Turn||Red Light Violation||
> |-|-|-|-|-|-|-|
> || ADR↑|EIR↑| ADR↑|EIR↑| ADR↑| EIR↑|
> |RGB|$92.8\pm0.8$|$85.5\pm0.5$|$67.3\pm1.0$|$79.7\pm1.1$|$66.1\pm1.4$|$83.0\pm1.3$|
> |LiDAR|$93.3\pm0.9$|$86.1\pm0.6$|$68.0\pm0.9$|$80.4\pm1.3$|$67.2\pm1.2$|$83.8\pm1.4$|
>
> >Performance comparison to late cooperation
>
> We conduct additional experiments using a late cooperation strategy for the CAV tasks. For each vehicle and each modality (RGB, LiDAR), we compute its decision output (action logits). Specifically, we use the same encoders as in CAML, ResNet-18 for RGB and PointTransformer for LiDAR, to extract features independently from each modality. These features are then passed through MLP prediction heads to generate action logits. To perform late cooperation, we first fuse the logits of the same modality from connected vehicles to the ego vehicle by averaging. Next, we fuse the averaged logits across modalities by taking their mean. The resulting fused multi-modal, multi-agent logits are used to determine the ego vehicle’s control actions for accident detection. We then apply the same knowledge distillation procedure as CAML, distilling from the RGB-LiDAR teacher model to a student model that only uses RGB at test time. The results are presented in the following table. As shown, CAML outperforms the late cooperation approach, as it enables the model to learn richer and more complementary representations across agents and modalities. In contrast, late cooperation only fuses output decisions, which may lead to information loss.
>
> |Approach| Overtaking|| Left Turn||Red Light Violation||
> |-|-|-|-|-|-|-|
> || ADR↑|EIR↑| ADR↑|EIR↑| ADR↑| EIR↑|
> |CAML|$92.8\pm0.8$|$85.5\pm0.5$|$67.3\pm1.0$|$79.7\pm1.1$|$66.1\pm1.4$|$83.0\pm1.3$|
> |Late Cooperation|$88.0\pm1.0$|$83.5\pm1.0$|$60.2\pm1.3$|$78.5\pm0.9$|$58.2\pm1.5$|$81.7\pm1.5$|
>
> >Difference between other RGB-based intermediate cooperation methods
>
> CAML is an intermediate cooperation approach. It has following advantages compared to existing RGB-based intermediate cooperation methods.
>
> 1. **Robustness under modality reduction**: Unlike typical RGB-based cooperation methods that assume consistent modality availability during both training and inference, CAML is explicitly designed to handle modality reduction at test time. It enables robust inference even when high-cost or unreliable modalities (e.g., LiDAR) are unavailable during deployment.
>
> 2. **Flexibility to agent and modality heterogeneity**: CAML is a general framework that supports a varying number of agents and heterogeneous modality availability per agent. This makes it particularly suited for resource-constrained environments where sensors may fail, communication may be intermittent, or hardware capabilities differ across agents.
>
> 3. **Cross-agent, cross-modal knowledge distillation**: CAML goes beyond traditional distillation by performing cross-agent, cross-modal knowledge transfer. It enhances the RGB representation by leveraging auxiliary modalities and collaborative knowledge shared among agents, leading to improved performance even in RGB-only inference settings.
>
> >Training complexity of pre-fusion CAML
>
> In terms of training complexity, Pre-fusion CAML is very similar to that of CAML, since both share a similar overall network architecture and use same modality-specific encoders. We illustrate in the table below with the collaborative semantic segmentation task.
>
> |Approach|Parameters|Time/epoch|
> |-|-|-|
> |AML|13.5M|3s|
> |CAML|25.5M|7s|
> |Pre-fusion CAML|25.7M|7s|
>
> Both CAML and its variant Pre-fusion CAML have their advantages, the preferable choice depends on the specific task requirements. If cross-agent modality alignment and consistency in feature representation are critical, CAML is more suitable. If preserving agent-specific contextual understanding and reducing inter-agent communication are priorities, Pre-fusion CAML may be preferred.

---

> > ### Comment · Reviewer_fUqC · 2025-08-02
> >
> > Thanks for the reply. The experimental results address my concerns.

---

> > > ### Author Response · Authors · 2025-08-05
> > >
> > > Dear reviewer, could you please raise the score, since we have addressed your concerns. We are deeply grateful for that!

---

> > > > ### Comment · Reviewer_fUqC · 2025-08-05
> > > >
> > > > I will change the score to relative positive.

---

> > > > > ### Author Response · Authors · 2025-08-05
> > > > >
> > > > > Thanks very much for your support! We appreciate your time in reviewing our work.

---

### Official Review · Reviewer_G7GG · 2025-06-24

**Clarity:** 3
**Significance:** 3
**Originality:** 2
**Rating:** 5
**Confidence:** 3

**Summary:**

This paper proposes a multi-agent system named CAML to address the problem of missing modalities at test time. It demonstrates a 58.1% improvement in accident detection performance and further validates the effectiveness of CAML in real-world scenarios.

**Questions:**

1. The paper evaluates only two modalities. Can the proposed framework generalize to settings with more modalities, and is it capable of handling arbitrary missing modalities at inference time?

2. In autonomous driving scenarios, when the ego vehicle has limited field of view, the V2V system aims to compensate by leveraging visual information from surrounding vehicles. This constitutes information-level missingness rather than modality-level. Can the authors clarify whether their method can handle such information-missing situations?

**Ethical Concerns:**

["NO or VERY MINOR ethics concerns only"]

**Final Justification:**

The experimental results in rebuttal address my concerns to a large extent.

**Limitations:**

yes

**Quality:**

3

**Strengths And Weaknesses:**

Strengths:

1. The paper presents a collaborative multi-agent system, enabling complementary information exchange between agents, which enhances the robustness of the system.

2. The effectiveness of CAML is validated on real-world data, further demonstrating its practical value for deployment.

Weaknesses:

1. The paper only evaluates the performance under missing LiDAR signals. However, the more informative setting would be to retain LiDAR and remove RGB input, which would better reflect the model’s robustness to missing modalities rather than reliance on a dominant one.

2. The paper does not mention existing work on missing modality problems, such as [1] [2], nor does it discuss the differences between CAML and these methods in terms of design or empirical performance.

[1] Ma M, Ren J, Zhao L, et al. Are multimodal transformers robust to missing modality?[C]//Proceedings of the IEEE/CVF conference on computer vision and pattern recognition. 2022: 18177-18186.
[2] Lee Y L, Tsai Y H, Chiu W C, et al. Multimodal prompting with missing modalities for visual recognition[C]//Proceedings of the IEEE/CVF Conference on Computer Vision and Pattern Recognition. 2023: 14943-14952.

---

> ### Author Rebuttal · Authors · 2025-07-30
>
> Thanks for your comments and feedback. We are pleased that you found our collaborative multi-agent design **effective in enabling complementary information exchange** and **enhancing system robustness**. We are also encouraged that you recognized the **practical value of CAML for deployment**, as **validated on real-world data**. Below, We address your concerns and will incorporate all feedback into the final version. We hope you'll consider raising the score.
>
> >Additional experiments of retaining LiDAR and removing RGB
>
> We conduct additional experiments by retaining LiDAR and removing RGB during testing for the experiments of collaborative decision-making for CAV. We present the results in the following table, with mean and std over four repeated runs. The performances of retaining RGB or LiDAR are similar but retaining LiDAR shows slightly better performance as LiDAR provides more precise spatial localization which is better for accident detection.
>
> |Modality during testing| Overtaking|| Left Turn||Red Light Violation||
> |-|-|-|-|-|-|-|
> || ADR↑|EIR↑| ADR↑|EIR↑| ADR↑| EIR↑|
> |RGB|$92.8\pm0.8$|$85.5\pm0.5$|$67.3\pm1.0$|$79.7\pm1.1$|$66.1\pm1.4$|$83.0\pm1.3$|
> |LiDAR|$93.3\pm0.9$|$86.1\pm0.6$|$68.0\pm0.9$|$80.4\pm1.3$|$67.2\pm1.2$|$83.8\pm1.4$|
>
> >Modalities
>
> Yes, the proposed framework can generalize to settings with more modalities, and it is capable of handling arbitrary missing modalities at inference time. Please note that we did not evaluate only two modalities, in the experiments of CAV, we evaluated RGB and LiDAR, in the experiments of aerial-ground robot collaboration, we also evaluated RGB and Depth.
>
> In addition, we conduct additional experiments, using three modalities for the CAV task, where each agent receives RGB, LiDAR, and state information (position and velocity) during training. At test time, only the RGB modality is available for each vehicle. As shown below, incorporating state information during training improves the system performance.
>
> |Modalities during training| Overtaking|| Left Turn||Red Light Violation||
> |-|-|-|-|-|-|-|
> || ADR↑|EIR↑| ADR↑|EIR↑| ADR↑| EIR↑|
> |RGB+LiDAR|$92.8\pm0.8$|$85.5\pm0.5$|$67.3\pm1.0$|$79.7\pm1.1$|$66.1\pm1.4$|$83.0\pm1.3$|
> |RGB+LiDAR+State|$94.0\pm0.7$|$86.4\pm0.6$|$69.7\pm1.2$|$81.7\pm1.3$|$68.3\pm1.2$|$84.7\pm1.0$|
>
> >Handling information-missing situations
>
> Yes, CAML can handle such information-missing situations. It is not only a multi-modal system but also a multi-agent framework. From the multi-modal perspective, CAML addresses missing modalities at the modality level. From the multi-agent collaboration perspective, CAML explicitly handles information-level missingness, such as when the ego vehicle has a limited field of view, by leveraging complementary information from surrounding vehicles through V2V communication. This enables the system to compensate for partial observations by integrating diverse views across agents, thereby enhancing perception and decision-making capabilities.
>
> >Discussion on related work
>
> Thanks for the suggestion. CAML differs from the two papers in following:
>
> 1. CAML leverages multi-agent collaboration and knowledge distillation to address both missing modality and partial agent coverage, which is not explored by the two referenced single-agent works.
>
> 2. The other two works focus on improving single-agent robustness to missing sensor inputs, either through transformer fusion strategies or prompting, without multi-agent collaboration or cross-agent information transfer.
>
> We will be sure to discuss these related works in the camera-ready version.

---

> > ### Comment · Reviewer_G7GG · 2025-08-02
> >
> > Thank you for your reply. The experimental results in rebuttal address my concerns to a large extent.

---

> > > ### Author Response · Authors · 2025-08-05
> > >
> > > Dear reviewer, could you please raise the score, since we have addressed your concerns. We are deeply grateful for that!

---

> > > > ### Comment · Reviewer_G7GG · 2025-08-05
> > > >
> > > > I will change the score to accept.

---

> > > > > ### Author Response · Authors · 2025-08-05
> > > > >
> > > > > Thanks very much for your support! We appreciate your time in reviewing our work.

---

### Official Review · Reviewer_eJXK · 2025-06-25

**Clarity:** 3
**Significance:** 2
**Originality:** 3
**Rating:** 4
**Confidence:** 3

**Summary:**

The paper introduces ​CAML, a novel framework for multi-agent systems that leverages multi-modal data during training (e.g., LiDAR + RGB) but operates with reduced modalities (e.g., RGB-only) during inference.

**Questions:**

1. The knowledge distrillation  has been proposed in AML, so what is the key innovation of this work?

2. How the knowledge is distilled from the teacher model to the student model is not clearly introduced.

3. There have been many collaborative perception frameworks. Can this method work in existing collaborative perception method?

**Ethical Concerns:**

["NO or VERY MINOR ethics concerns only"]

**Final Justification:**

4: Borderline accept: Technically solid paper where reasons to accept outweigh reasons to reject, e.g., limited evaluation. Please use sparingly.

**Limitations:**

As disscussed in the questions.

**Quality:**

3

**Strengths And Weaknesses:**

Strength:

1. ​Multi-agent knowledge distillation: A teacher model (full modalities) distills knowledge into a student model (reduced modalities), preserving collaborative insights.
​
2. Flexible modality/agent configurations: Supports varying numbers of agents/modalities between training and testing.

​3. Strong empirical results: 58.1% higher accident detection in autonomous driving and 10.6% mIoU gain in aerial-ground robot segmentation versus baselines.

Weakness:

1. Technical innovation: It seems that the paper is just an extension from the single-agent AML method to the multi-agent CAML scenario.

2. How the knowledge is distrillated from teacher model to the student model is not clearly introduced.

3.  The performance difference from AML to CAML is very limited.

---

> ### Author Rebuttal · Authors · 2025-07-30
>
> Thanks for your comments and feedback. We are glad that you found our approach **preserving collaborative insights, flexible for modality/agent, with strong empirical results**. Below we answer your questions and will incorporate all feedback to the final version. We hope you'll consider raising the score.
>
> >Key innovation
>
> Please note that our proposed CAML is not merely an extension of AML. Multi-agent settings introduces substantial complexity, requiring careful consideration of new challenges. CAML specifically addresses the unique challenges of multi-agent systems, including heterogeneity in sensing modalities, dynamic team compositions, and team-level decision-making. CAML has broader applications of connected autonomous driving, collaborative search and rescue, as well as intelligent transportation. To the best of our knowledge, this work is the first to propose a flexible and principled framework capable of tackling a broad range of collaborative tasks in multi-agent systems where limited modalities are available for inference. Unlike prior work that either focuses on multi-agent collaboration but without addressing missing modality at test time, or explores AML ideas solely in single-agent settings, CAML unifies these directions for the first time.
>
> >Knowledge distillation
>
> The knowledge is distilled from the teacher to the student by having the student mimic the teacher’s behavior. For example, in the CAV experiments, we first train a teacher model offline using a cross-entropy loss, where each vehicle has access to both RGB and LiDAR data. Then, we train a student model that receives only RGB input, learning to replicate the behavior of the teacher model. For each data sample, the student receives the same RGB image as the teacher. The training loss for the student model combines two terms: (1) a distillation loss using KL divergence between the student and teacher outputs, which encourages the student to closely match the teacher’s output, and (2) a task loss using cross-entropy between the student’s prediction and the ground-truth label. We provided these details in Appendix A.5 in paper.
>
> >Performance difference from AML to CAML
>
> Please note that the performance improvement of CAML over AML is not marginal, but substantial. In collaborative decision-making for CAV, CAML achieves up to a 58.1% improvement in accident detection compared to AML. Additionally, in real-world aerial-ground robot experiments for collaborative semantic segmentation, CAML achieves up to a 10.6% improvement in mIoU over AML. These results highlight the significant advantages of CAML compared to AML.
>
> >Working in existing collaborative perception method
>
> Yes, CAML can work with existing collaborative perception methods. It is a general framework that is not limited to specific modalities and can flexibly incorporate different types of sensory inputs. CAML is particularly well-suited for resource-constrained environments where certain modalities may be missing during inference. By enabling effective reduced-modality inference, CAML helps reduce computational cost and improve system robustness during inference, otherwise the system may not work due to missing modality.

---

> > ### Comment · Reviewer_eJXK · 2025-08-05
> > **Thanks for the responses**
> >
> > Regarding the novelty, even if it is the first time that the AML idea is explored in a multi-agent scenario, the key technical innovations are still limited by transferring the knowledge distilling method from the single-agent case to the multi-agent case.
> >
> > Regarding knowledge distillation. I am clear with the pipeline, but haven't seen in-depth studies about why knowledge distillation is very useful in CAML.

---

> ### Author Response · Authors · 2025-08-05
>
> We would like to clarify that our technical innovations are not limited by transferring knowledge distillation from single-agent to multi-agent settings.
>
> First, we also introduce technical innovations including: 1) System design, a collaborative teacher-student architecture tailored to the multi-agent setting. 2) Cross-agent, cross-modal knowledge transfer that extends beyond traditional KD, enabling better handling of partial observability and supporting team-level coordination. 3) Efficient runtime design: as shown in Table 1 in paper, our model achieves much lower communication bandwidth and inference latency than prior collaborative methods.
>
> Second, our use of KD is motivated by practical deployment constraints of multi-agent systems, where certain modalities may be unavailable at test time due to missing data or resource limitations. To address this, we leverage auxiliary modalities during training and distill their information into a student model for robust and efficient inference. This goes well beyond simply applying KD from the single-agent context.
>
> Third, multi-agent systems pose unique challenges such as heterogeneity in sensing modalities, dynamic team compositions, and team-level decision-making. Transitioning from single-agent to multi-agent learning is fundamentally non-trivial, multi-agent system learning or Agentic AI are established and complex research areas in their own right.
>
> The usefulness of knowledge distillation in CAML is verified through extensive and in-depth experiments. We distill knowledge from a powerful teacher model into a student model, so that the student model learns to handle missing modalities at test time and operate efficiently in resource-constrained environments. This is demonstrated across diverse scenarios, including collaborative decision-making in CAV and aerial-ground collaborative segmentation, using multiple modalities such as RGB, LiDAR, and Depth, please see Fig. 3–5 and Tables 1–3 in paper. We further validate this by conducting additional experiments incorporating state information as a third modality, showing that distilling state information during training improves the system performance, as presented in the following table.
>
> |Modalities during training| Overtaking|| Left Turn||Red Light Violation||
> |-|-|-|-|-|-|-|
> || ADR↑|EIR↑| ADR↑|EIR↑| ADR↑| EIR↑|
> |RGB+LiDAR|$92.8\pm0.8$|$85.5\pm0.5$|$67.3\pm1.0$|$79.7\pm1.1$|$66.1\pm1.4$|$83.0\pm1.3$|
> |RGB+LiDAR+State|$94.0\pm0.7$|$86.4\pm0.6$|$69.7\pm1.2$|$81.7\pm1.3$|$68.3\pm1.2$|$84.7\pm1.0$|

---

### Official Review · Reviewer_yoA6 · 2025-07-01

**Clarity:** 3
**Significance:** 2
**Originality:** 2
**Rating:** 4
**Confidence:** 3

**Summary:**

The paper introduces CAML, a framework that integrates auxiliary modality learning with multi-agent collaboration. The core idea is to enable agents to share multi-modal data during training and still function effectively at inference time with reduced modality input. This is particularly useful for real-world applications like connected autonomous vehicles and collaborative robotics, where some modalities may be expensive or unavailable during deployment. CAML leverages knowledge distillation  to transfer information from a full-modality teacher model to a reduced-modality student model.

**Questions:**

In light of the previously mentioned weaknesses:

- Can the authors include a brief experiment showing how CAML handles more than 2 modalities? and whether it is possible to handle modalities dynamically at test time?

- Can the authors clarify whether CAML learns modality preferences implicitly, and if so, how it would behave if modalities become available intermittently?

**Ethical Concerns:**

["NO or VERY MINOR ethics concerns only"]

**Final Justification:**

The authors provided thorough explanations and additional results that addressed my concerns to a significant extent. While theoretical scalability to many modalities and their informativeness remains unclear, the practical constraints based on benchmarks are acknowledged. I appreciate the clarifications and encourage the authors to include these points and limitations in the manuscript.

**Limitations:**

- Can the authors add a short discussion in the limitations section about how CAML might behave in cases where all available modalities are uninformative or corrupted (e.g., in bad weather)? Could the system detect such cases or be extended to estimate uncertainty?

**Paper Formatting Concerns:**

There are no formatting concerns.

**Quality:**

3

**Strengths And Weaknesses:**

Strengths:

- The paper is clearly written and well-structured. The authors effectively motivate the problem and identify key limitations of existing approaches.
- The proposed architecture is conceptually sound, and its components are well-justified. The support for both centralized and decentralized communication setups is clearly discussed.
- The authors provide a complexity analysis of the approach in both centralized and decentralized settings, which adds clarity to the framework’s practical implications.

Weaknesses:

- While the framework is well described and motivated, the evaluation could benefit from a broader experimental setup to more thoroughly support the authors’ claims. Specifically, the evaluation relies only on simulated data and scenarios using the AUTOCASTSIM benchmark, which may not fully capture the complexity of real-world environments. Experiments in real-world settings, using datasets such as RCooper [1], would help validate the approach more robustly.

- The setup assumes that modality availability remains static at inference time, and only two modalities (RGB and LiDAR) are considered. This raises several questions: How would the model perform under dynamically varying modalities at test time? For instance, if RGB is intermittently unavailable while LiDAR becomes available, how does the system adapt? Would it prioritize one modality over another, and does the model learn such preferences?

- The evaluation would also benefit from an extension to other tasks, such as collaborative tracking, as this would enable more direct comparisons with additional baselines like CoBEVT [2], F-Cooper[3], and Where2Comm [4].

- Related to the previous points: The paper does not discuss scenarios where all available modalities might be uninformative or corrupted (e.g., in adverse weather conditions). A short discussion on whether the system could support uncertainty estimation in modality effectivenes, or at least detect such failure cases, would be a valuable addition to the limitation section of the paper.

References:

[1]  R. Hao et al., "RCooper: A Real-world Large-scale Dataset for Roadside Cooperative Perception," in CVPR, 2024.

[2] R. Xu et al., "CoBEVT: Cooperative Bird’s Eye View Semantic Segmentation with Sparse Transformers," in CoRL, 2022.

[3] Q. Chen et al., "F-Cooper: Feature-Based Cooperative Perception for Autonomous Vehicles Using 3D Point Clouds," in Proc. ACM/IEEE SEC, 2019.

[4] Y. Hu et al., "Where2comm: Communication-Efficient Collaborative Perception via Spatial Confidence Maps," NeurIPS 2022.

---

> ### Author Rebuttal · Authors · 2025-07-30
>
> Thanks for your comments and feedback. We’re glad you found our paper **well-written**, **well-motivated**, and the **architecture sound** with **clear discussion of both communication setups and complexity analysis**. Below we address your concerns and will incorporate all feedback in the final version. We hope you'll consider raising the score.
>
> >Experimental evaluation
>
> We conduct additional experiments as requested, using three modalities for the CAV task, where each agent receives RGB, LiDAR, and state information (position and velocity) during training. At test time,  only the RGB modality is available for each vehicle. We show the results in the following table, with mean and std over four repeated runs. As we can see, having state information as another modality improves the system performance. This setup further demonstrates CAML’s ability to leverage richer multi-modal data during training while maintaining effective performance under reduced-modality conditions at inference.
>
> |Modalities during training| Overtaking|| Left Turn||Red Light Violation||
> |-|-|-|-|-|-|-|
> || ADR↑|EIR↑| ADR↑|EIR↑| ADR↑| EIR↑|
> |RGB+LiDAR|$92.8\pm0.8$|$85.5\pm0.5$|$67.3\pm1.0$|$79.7\pm1.1$|$66.1\pm1.4$|$83.0\pm1.3$|
> |RGB+LiDAR+State|$94.0\pm0.7$|$86.4\pm0.6$|$69.7\pm1.2$|$81.7\pm1.3$|$68.3\pm1.2$|$84.7\pm1.0$|
>
> Please note that our evaluation does not only rely on simulation data and scenarios using the AUTOCASTSIM benchmark, which is used for the experiments of the connected autonomous vehicles (CAV). In addition, we evaluated on aerial-ground robot collaboration with real-world dataset CoPeD for collaborative semantic segmentation. This verifies the effectiveness of our approach in real-world environments.
>
> Thank you for the suggestions to evaluate on tasks such as collaborative tracking and to explore datasets like RCooper. We agree that these directions are valuable and complementary to our work, but incorporating such additional experiments within the very limited rebuttal period is not feasible. Moreover, our collaborative decision-making experiments for CAV actually already implicitly involve aspects of collaborative tracking, as vehicles must first be localized and tracked before making decisions. We appreciate the suggestions and will be sure to include discussions of these potential extensions in the future work section of the final version.
>
> >Modalities
>
> We conduct additional experiments including state information (position and velocity) as another modality for the CAV tasks, as shown in the above table. Moreover, our work does not only consider RGB and LiDAR modalities. In the aerial-ground robot experiments, we incorporate Depth as a different modality. These evaluations further demonstrate CAML’s flexibility in handling diverse sensory inputs across different tasks and environments.
>
> Yes, it is possible to handle modalities dynamically at test time or when modalities become available intermittently. CAML does not learn modality preferences implicitly, it can utilize all modalities available at time time. In the student model of CAML, the modalities for each agent can be configured individually. The agents do not all need to have the same modalities. Although the current student model training does not dynamically change the modalities over time. As noted in the future work section, we can handle dynamic changes without changing the architecture. Our planned solution is to incorporate a modality dropout strategy. During training of knowledge distillation, we can randomly drop out or mask some modality for each agent in the student model. This allows the student model to learn robustness to dynamic modality availability and adapt effectively over time.
>
> >Discussion in limitation section
>
> In extreme scenarios where all available modalities are uninformative (e.g., due to adverse weather), system performance may degrade. CAML improves robustness by leveraging auxiliary modalities and multi-agent collaboration during training. Additionally, our system can support uncertainty estimation to access modality effectiveness. CAML can be extended to incorporate uncertainty estimation mechanisms to further improve system robustness when available modalities are noisy. This would allow the system to detect and better handle such cases. We will include these discussions into the limitations section of the final version.

---

> > ### Comment · Reviewer_yoA6 · 2025-08-05
> >
> > I thank the authors for their thorough response, detailed explanations, and the additional results provided, which have addressed my concerns to a certain extent. While it remains unclear how many modalities CAML can theoretically handle, due to the lack of a fundamental analysis in the paper, I acknowledge that, in practice, the number of modalities is likely constrained by the benchmarks used.
> > I encourage the authors to incorporate these additional explanations into the manuscript and to clearly highlight the potential limitations of their approach in this regard. Based on their response, I will raise my score.

---

> > > ### Author Response · Authors · 2025-08-05
> > >
> > > Thanks very much for your support! We will be sure to incorporate these additional explanations and experimental results into the camera-ready version and to clearly highlight the potential limitations of the approach. We appreciate your time in reviewing our work.

---

### Note · Authors · 2025-08-14

We sincerely thank all reviewers for their positive assessments and constructive engagement, which further strengthened our work. We appreciate the recognition that *the paper is clearly written and well-structured, with strong problem motivation, a conceptually sound and well-justified architecture*, and *clear support for both centralized and decentralized communication along with practical complexity analysis*. We are grateful that reviewers highlighted CAML’s *flexibility across agents and modalities configurations*. Empirically, CAML achieves strong results, including a 58.1% improvement in accident detection and a 10.6% mIoU gain in aerial-ground robot segmentation, validated on real-world data, demonstrating its technical soundness and practical effectiveness.

During the rebuttal, we conduct additional experiments, including having three modalities for the connected autonomous vehicles (CAV) task, retaining LiDAR and removing RGB during testing, and comparing CAML with the late cooperation approach. We also clarify contributions beyond AML: a collaborative teacher-student design tailored to **multi-agent settings, cross-agent and cross-modal knowledge transfer** for ***partial observability and team-level coordination***, and an efficient runtime design with *reduced bandwidth and latency* compared to prior cooperative methods. **CAML's multi-agent settings introduce substantial complexity, which CAML addresses by handling heterogeneous sensing modalities, dynamic team compositions, and team-level decision-making**. We further highlight differences from other RGB-based intermediate cooperation methods, including *robustness under modality reduction, flexibility to agent and modality heterogeneity, and cross-agent, cross-modal knowledge distillation*.

We will incorporate all additional experimental results, explanations and discussions into the camera-ready version.  We would like to sincerely thank all reviewers again for the helpful suggestions that notably enhance the exposition of the paper.

---

### Decision · Program_Chairs · 2025-09-17

**Decision:**

Accept (poster)

**Comment:**

This paper considers the missing-modality problem for multi-modal collaborative multi-agent systems. The approach centers on a knowledge distillation method, where a full-modality teacher model is distilled into a reduced-modality student model, extending auxiliary modality learning methods. Strong results are shown on challenging accident-prone scenarios where agents can have limited visibility. The reviewers found the paper well-motivated and written, and appreciated the strong performance on real-world data and ablation/complexity analysis. Several concerns regarding clarification of the contribution (especially compared to AML), having more thorough sets of modality experiments (e.g. no RGB), and clarification of the training complexity. The authors provided a strong rebuttal, including with new results answering some of these questions (results of additional modalities, training complexity results, etc.), and all reviewers were satisfied with the paper. I recommend acceptance, and for the authors to incorporate the clarifications and results in the final paper.